# Association between Androgen Deprivation Therapy and the Risk of Inflammatory Rheumatic Diseases in Men with Prostate Cancer: Nationwide Cohort Study in Lithuania

**DOI:** 10.3390/jcm11072039

**Published:** 2022-04-05

**Authors:** Mingaile Drevinskaite, Jolanta Dadoniene, Dalia Miltiniene, Ausvydas Patasius, Giedre Smailyte

**Affiliations:** 1Laboratory of Cancer Epidemiology, National Cancer Institute, P. Baublio 3B, LT-08406 Vilnius, Lithuania; ausvydas.patasius@nvi.lt (A.P.); giedre.smailyte@nvi.lt (G.S.); 2State Research Institute Centre for Innovative Medicine, LT-08410 Vilnius, Lithuania; jolanta.dadoniene@mf.vu.lt (J.D.); dalia.miltiniene@santa.lt (D.M.); 3Department of Public Health, Institute of Health Sciences, Faculty of Medicine, LT- 03101 Vilnius, Lithuania; 4Clinic of Rheumatology, Orthopaedics Traumatology and Reconstructive Surgery, Institute of Clinical Medicine, Vilnius University Faculty of Medicine, LT-08410 Vilnius, Lithuania

**Keywords:** androgen deprivation therapy, inflammatory rheumatic disease, rheumatoid arthritis, prostate cancer, adverse effect

## Abstract

Background: The aim of this study was to assess the association between androgen deprivation therapy (ADT) and the risk of inflammatory rheumatic diseases in men with prostate cancer. Methods: Patients with prostate cancer between 2012 and 2016 were identified from the Lithuanian Cancer Registry and the National Health Insurance Fund database, on the basis of rheumatic diseases diagnoses and information on prescriptions for androgen deprivation therapy. Cox proportional hazard models were used to estimate hazard ratios (HR) to compare the risks of rheumatic diseases caused by androgen deprivation therapy exposure in groups of prostate cancer patients. Results: A total of 12,505 prostate cancer patients were included in this study, out of whom 3070 were ADT users and 9390 were ADT non-users. We observed a higher risk of rheumatic diseases in the cohort of prostate cancer patients treated with ADT compared with ADT non-users (HR 1.55, 95% confidence interval (CI) 1.01–2.28). Detailed risk by cumulative use of ADT was performed for rheumatoid arthritis, and a statistically significant higher risk was found in the group with longest cumulative ADT exposure (>105 weeks) (HR 3.18, 95% CI 1.39–7.29). Conclusions: Our study suggests that ADT usage could be associated with increased risk of rheumatoid arthritis, adding to the many known side effects of ADT.

## 1. Introduction

Prostate cancer is one of the most prevalent malignant neoplasms and is the second most common cause of cancer-related deaths in men worldwide [1]. Androgen deprivation therapy (ADT) has been a backbone therapy for high-risk and certain-intermediate-risk localized, locally advanced, and metastatic prostate cancers [2]. ADT can be administered by surgery (surgical castration) or medication (gonadotropin-releasing hormone (GnRH) agonists, GnRH antagonists, oral antiandrogens, or a combination of GnRH analogues and antiandrogens in some clinical cases) [3]. Despite its proven survival benefit, hypogonadism, which is produced by ADT, leads to several adverse effects, such as atherosclerosis, type 2 diabetes, obesity, arterial hypertension, anemia, impotence, reduced quality of life, cognitive dysfunction and psychiatric effects, reduced bone mineral density, and increased risk of cardiovascular diseases [4,5,6,7,8]. 

By inhibiting testicular androgen secretion and androgen receptors, ADT rapidly decreases serum concentrations of testosterone to castration level. Testosterone is believed to have a suppressive role on multiple aspects of the immune system, which may partly explain why women are prone to autoimmune diseases more than men [9]. Since testosterone has anti-inflammatory properties, the decline of testosterone while using ADT supports the proinflammatory process, for example, by activating the T cell response in organs of the immune system [10]. It is well known that ADT causes a variety of adverse effects; however, its link to inflammatory RD is not clearly established. A first case report series of three patients raised a plausible question of whether ADT could be associated with the development of rheumatoid arthritis (RA) [11]. However, to date, information is very limited. To our knowledge, only three observational studies have analyzed inflammatory rheumatic diseases (RD) risk in ADT patients, and the reported findings were conflicting [12,13,14].

The aim of this analysis was to evaluate whether ADT use could be associated with RD risk in Lithuanian patients with prostate cancer.

## 2. Materials and Methods

### 2.1. Study Design and Population

A retrospective cohort design was used to examine the relationship between ADT use and risk of RD. For this study, from the Lithuanian Cancer Registry database, we identified a cohort of men aged 40–79 years with primary prostate cancer (ICD–10 code C61), diagnosed between 1 January 2012 and 31 December 2016. 

All prostate cancer cases were linked to the National Health Insurance Fund (NHIF) database in order to obtain information regarding the diagnosis of RD and information on prescriptions of antiandrogens and GnRH agonists. The link between different databases was based on personal identification codes, which are unique to each resident of Lithuania.

The group of RDs included rheumatoid arthritis (ICD–10 codes M05, M06), spondyloarthropathies (M07, M45, M46), and systemic connective tissue diseases (M30–M35). Because the diagnoses causing admission are logged into the database by healthcare providers, to increase the sensitivity of case definition for RD, only patients who had received prescriptions for reimbursed medication were treated as RD cases. Medications for RD treatment include glucocorticoids (prednisone or methylprednisolone), conventional synthetic DMARDs (methotrexate, azathioprine, leflunomide, sulfasalazine, and hydrochloroquine), or biologic DMARDs (infliximab, etanercept, adalimumab, tocilizumab, or rituximab with available biosimilars). Patients with a first RD diagnosis in the NHIF database in 2012 included prevalent cases of RD.

ADT users were identified by reimbursed use of GnRH agonists (triptorelin, leuprolelin, histrelin, goserelin) and antiandrogens (bicalutamide, cyproterone acetate). Out of 13,697 patients, 3915 were GnRH-only users and 87 were antiandrogens-only users. After the link with RD data, prostate cancer patients with RD diagnosis in 2012 (127 cases) and patients with RD diagnosis before cancer diagnosis (87 cases) were excluded from the study cohort. We also excluded those cancer patients from the analysis whose first ADT prescription was more than 90 days after prostate cancer diagnosis (702 cases) and those identified as death-certified patients (276 cases). A total of 45 prostate cancer patients were treated with antiandrogens only; they were excluded from the analysis to make the study group homogenous (due to the smaller effect of androgen deprivation). 

Finally, 12,460 prostate cancer patients were included in the study cohort. Identified patients were followed either until the date of RD diagnosis, until 31 December 2019, or until the date of death, whichever came first.

### 2.2. Statistical Analysis

We analyzed the risk of RD between two groups of prostate cancer patients (ADT users and ADT non-users). In order to evaluate RD incidence by ADT use, we calculated the exact person years at risk for each patient from the date of diagnosis till the end of the follow-up period.

Cox proportional hazard models were used to estimate hazard ratios and their 95% confidence intervals to compare the risk of RD in groups of prostate cancer patients according to ADT exposure. Multivariate adjusted Cox proportional hazard models, including age and stage at diagnosis, were conducted to estimate the effect of ADT on RD risk. Association between duration of GnRH agonists use and RD risk was assessed by dividing cumulative duration of use into the following intervals: 4–43 weeks, 44–104 weeks, and more than 105 weeks. 

All statistical analyses were carried out using STATA statistical software (version 15.1; College Station, TX, USA). The Vilnius Regional Biomedical Research Ethics Committee approved this study (approval number 158200-17-958-462).

## 3. Results

Among the 12,505 prostate cancer patients, there were 3070 ADT users and 9390 ADT non-users. The mean follow-up time for ADT users and non-users was 1624.79 and 1859.09 days, respectively. The mean age for ADT users was 68 years, and it was higher than ADT non-users (63 years). Study group characteristics are shown in Table 1. 

During the study period, 133 RD cases were diagnosed in the group of prostate cancer patients (incidence rate 2.16 per 1000 men). A total of 97 cases of RD were diagnosed among ADT non-users, giving an incidence rate of 2.03 per 1000 men, while among ADT users, the incidence was higher (36 cases: 2.64 per 1000).

There was a higher risk of RD in the cohort of patients diagnosed with prostate cancer and treated with ADT compared with ADT non-users (Table 2). The unadjusted Cox regression analysis did not show a statistically significant higher risk of RD among ADT users (HR 1.24, 95% CI 0.85–1.82). Despite the positive correlation that did not reach statistical significance in the unadjusted analysis, it became significant after adjusting for age and stage (HR 1.55, 95% CI 1.01–2.28).

Risks for the specific RD were also assessed for ADT users compared with the ADT non-users (Table 2). There was a statistically significant higher risk of RA among ADT users (ICD–10 codes M05, M06) (HR 2.20, 95% CI 1.27–3.82), and it remained higher after adjustment for stage and age (HR 2.53, 95% CI 1.42–4.52). 

Detailed risk by cumulative use of ADT was performed for RA only. After adjustment for age and stage, a statistically non-significant higher risk for RA was found in the group of ADT users with shortest duration of exposure (Table 3). A statistically significant higher risk of RA was observed for other groups of ADT users and the risk was highest in the group with the longest cumulative exposure (HR 3.18, 95% CI 1.39–7.29). 

## 4. Discussion

The main finding of this study was that there is a higher incidence and risk of RD in a male population diagnosed with prostate cancer and treated with ADT compared with the ADT non-users. Disease-specific risk analysis showed a higher risk for RA only. Risk of RA was increasing with increasing cumulative ADT use. 

There is a clearly established relation between Sjogren syndrome and lymphoma [15]. RA has also been linked to lymphoma and lung carcinoma, and it was observed that lymphomas and lung malignancies are more frequent in RA patients compared with the general population [16,17].

Multiple interactions exist between malignancies and rheumatic diseases. One major topic which is of concern for RD patients is their susceptibility to the development of tumors of different origins, assuming that permanently sustained inflammation is responsible for transition into cancer. However, there is also research suggesting that RD development in cancer patients has a relation mostly to sex hormone deprivation therapies, such as aromatase inhibitors for breast cancer, antiandrogens for prostate cancer, and most recently, checkpoint inhibitors used in melanoma, urothelial cancer, and kidney cancer [18,19,20].

Generally, we link androgen suppression with a decrease in testosterone and its precursors’ level as the main mechanism of ADT action. Since testosterone and its precursors (dehydroepiandrosterone and dehydroepiandrosterone-s) have anti-inflammatory properties, the decline of these hormones supports the proinflammatory process. The proinflammatory process is established by activating the T cell response in the organs of the immune system, though data about these mechanisms are scarce [10].

It is well known that sex hormone deprivation therapies, such as antiandrogens, may induce bone loss and lead to osteoporosis; however, its link to inflammatory RD is not clearly established, and some contradicting observations have been published recently [21]. To our knowledge, there are three observational studies and one case report series where plausible ADT and RA associations have been addressed [11,12,13,14].

In our study, we observed a statistically significant increased risk of all RDs in the ADT users compared with ADT non-users (HR 1.55, 95% CI 1.01–2.28 and 2.53, 95% CI 1.42–4.52) and of RA patients specifically (HR 2.20, 95% CI 1.34–4.80). Similar results to ours were found in a study by Yang et al., where ADT usage was associated with a 23% increased risk of being diagnosed with RA (HR 1.23, 95% CI 1.09–1.40) in men aged 66 years or older. Moreover, the risk of being diagnosed with RA increased with a longer duration of ADT, from 19% within 1–6 months (HR 1.19, 95% CI 1.06–1.33) to 33% ≧ 13 months (HR 1.33, 95% CI 1.13–1.56) [12]. In our study, analysis of the cumulative use of ADT showed an increasing risk of RA, and the risk was highest in the group with the longest cumulative exposure (HR 3.18, 95% CI 1.39–7.29). 

A population-based cohort study by Klil-Drori et al. included more than 30,000 men with prostate cancer, among whom 63 patients were newly diagnosed with RA. The authors observed no association between ADT and increased risk of RA compared with ADT non-users (HR 0.84, 95% CI 0.49–1.45). In addition, the results did not differ in terms of duration of use or ADT type [14].

On the contrary, another population-based, nationwide cohort study conducted by Liu et al.—which used both a propensity-score-matched analysis and multivariable regression models to examine whether ADT is associated with autoimmune diseases—observed that ADT usage significantly decreased the risk of all individuals in the study, including autoimmune diseases (HR 0.62, 95% CI 0.51–0.75) [13]. The risk of developing RA while using ADT was 38% lower compared with ADT non-users (HR 0.62, 95% CI 0.38–1.03). Interestingly, a longer duration of ADT use was associated with a decrease in risk of autoimmune diseases (<12 months of using ADT (HR 0.67, 95% CI 0.49–0.90) vs. ≧ 12 months ADT use (HR 0.61, 95% CI 0.49–0.77). They hypothesized that long-term use of ADT may cause the adaptation of the immune response [13]. In our study, we did not find a statistically significant increase in risk in terms of other rheumatoid diseases, except for RA. 

Our findings relate specifically to rheumatoid arthritis, and not to systemic lupus erythematosus or ankylosing spondylitis. It is generally accepted that inflammation measured with C-reactive protein is associated with rheumatoid arthritis more often than with the two other RDs investigated in this study. Moreover, inflammation is driven by T cells as a major player in pathogenesis of RA. T cells entering synovial tissue may cause the clinical features of rheumatoid arthritis, while systemic lupus erythematosus and ankylosing spondylitis are not primarily related to T cell activation and synovium as a targeted tissue. According to the androgen suppression demonstrated in castrated mice models, the increase in circulating T cells from the thymus may occur with inflammation following the release of cytokines [10]. The study also demonstrated that castration enhanced thymic regeneration in young male mice, and an increase in thymic-derived T cells was observed in elderly male humans. 

The strengths of our study include the large cohort size, the population-based design, and the use of real-world data. Our study has several limitations. First, we could not account for the known delays associated with RD diagnosis. Secondly, it was impossible to adjust for the established risk factors, such as socioeconomic status, education, or smoking. Finally, this population-based study was a retrospective study, based on a small number of events and a relatively short observation period. Therefore, our results should be interpreted with caution. Further retrospective and prospective studies are needed to evaluate the ADT relation not only with RA, but also with other autoimmune diseases, because this association may be of clinical importance.

## 5. Conclusions

Our study suggests that ADT use could be associated with an increased risk of rheumatoid arthritis, a possible addition to the many presently known side effects of ADT.

## Figures and Tables

**Table 1 jcm-11-02039-t001:** Baseline characteristics of men with prostate cancer, separated according to ADT use.

	All Patients	ADT Users	ADT Non-Users	*p* Value
N (%)	12,460 (100)	3070 (24.64)	9390 (75.36)	
Mean follow up time, years (SD)	4.93 (1.91)	4.45 (2.17)	5.09 (1.78)	<0.001
Mean age at diagnosis, years (SD)	64.62 (7.46)	68.21 (6.98)	63.45 (7.23)	<0.001
Stage				
Localized (%)	6471 (51.93)	828 (26.97)	5643 (60.10)	1.00
Locally advanced (%)	1921 (15.42)	1157 (37.69)	764 (8.14)	
Distant (%)	349 (2.80)	266 (8.66)	83 (0.88)	
Unknown (%)	3719 (29.85)	819 (26.68)	2900 (30.88)	

**Table 2 jcm-11-02039-t002:** Hazard ratios (HR) for RD in prostate cancer patients according to use of ADT and specific RD.

	Events	HR	95% CI	*p* Value	aHR *	95% CI	*p* Value
ADT free cohort		ref.	-		ref.		
ADT users	36	1.24	0.85–1.82	0.268	1.55	1.01–2.28	0.046
Rheumatoid arthritis (ICD–10 codes M05, M06),	21	2.20	1.34–4.80	0.005	2.53	1.42–4.52	0.002
Spondyloarthropathies (M07, M45, M46)	9	0.76	0.37–1.57	0.465	0.92	0.44–1.96	0.838
Systemic connective tissue diseases (M30–M35).	6	0.78	0.54–1.90	0.587	0.77	0.31–1.93	0.575

* Adjusted for age and stage.

**Table 3 jcm-11-02039-t003:** Hazard ratios (HR) for RA in men with prostate cancer on GnRH agonists according to cumulative duration of exposure.

Duration of ADT Exposure	Events	HR	95% CI	*p* Value	aHR *	95% CI	*p* Value
ADT free cohort		ref.	-		ref.		
4–43 weeks	4	1.37	0.48–3.89	0.548	1.59	0.54–4.64	0.400
44–104 weeks	8	2.46	1.13–5.34	0.023	2.89	1.25–6.68	0.013
>105 weeks	9	2.67	1.27–5.59	0.009	3.18	1.39–7.29	0.006

* Adjusted for age and stage.

## Data Availability

Data is available upon reasonable request.

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
