# Peer review of "Association between Androgen Deprivation Therapy and the Risk of Inflammatory Rheumatic Diseases in Men with Prostate Cancer: Nationwide Cohort Study in Lithuania"

_jcm, 2022, doi:10.3390/jcm11072039_

Round 1
Reviewer 1 Report
The article topic is original, it definitely brings new knowledge to the area and opens a field of research. The study suggests how the use of ADT could be associated with the development of rheumatoid arthritis in patients with prostate cancer.
The results are significant, however, not conclusive. Further studies should be performed, the authors cannot have any conclusive findings yet, but as they stated in their article, improvements could be done to straighten their research.
The article is well presented, the reading is easy and accessible. The tables are well presented and described. The methodology used, the selection of the data, including the strengths and limitations are exposed by the authors.
The references are little, however, not much previous information is available. Only once self-citation was reported, which justify the group work and interests.
Author Response
Authors are grateful for evaluation of our paper and comments.
Reviewer 2 Report
The prostate cancer (PCa) almost occurs in elders. I am not sure the authors how to adjusted age from chronic inflammatory diseases and PCa-related rheumatic diseases. In the introduction, the authors just provided the rare refs. of androgen associated RA and mentioned the information is very limited. But, this seems not truth according to literature survey. The investigative motivation is weak, and the statistical tables are the reduced results. The authors must present the detail statistical analysis procedure and the criteria of adjustment for age and stage.
Author Response
We do not agree with this evaluation of our manuscript. This is retrospective observational cohort study. In this study we used standard method to analyse association between ADT exposure and RD risk in patients with prostate cancer (Cox proportional hazard models). A Cox model is a well-recognized statistical technique for exploring the relationship between the survival of a patient and several explanatory variables. A Cox model provides an estimate of the treatment effect on survival (or risk of other disease, or death) after adjustment for other explanatory variables It allows us to estimate the hazard (or risk) of death, or other event of interest, for individuals, given their prognostic variables. In our study multivariate adjusted Cox proportional hazards models included age and stage at diagnosis. Also, to date only three observational studies have analysed inflammatory rheumatic diseases RD risk in ADT patients. It is mentioned in lines 57-59 of the manuscript.
Reviewer 3 Report
1- Repetition in line 44
2- Why exclude patients with first ADT prescription more than 90 days after prostate cancer diagnosis (702 cases). These are patients who have prostate cancer and who took long term ADT, I think they shouldn't be excluded.
3- Why include patients who take anti androgen just to exclude them later on. you could have searched only for patients using LHRH agonists/antagonists.
4- Inclusion and exclusion criterias should be listed clearly.
5- Could you provide a brief explanation to how and why the duration intervals of ADT were chosen? and why the intervals don't include the 40-44 weeks interval? A more suitable interval division would be to make the cut based on different indication of ADT use ( like 6 months for intermediate risk groups with radiotherapy, up to 2 years for high risk, and more than 2 years for metastatic...)
6- Many phrases should be restructured and simplified like line 85, 96-99
7- T-test should be used to compare Age and duration of follow-up , Anova test should be used to compare the different stages between users and non-users. P values (and F values for Anova) are needed to assess the statistical significance between the 2 groups.
8- Line 121: in the unadjusted regression, the correlation is not statistically significant because the confidence interval of the HR goes lower than 1.
the Authors could mention that the correlation being positive without reaching statistical significance in the unadjusted analysis, which become significant after adjusting for age and stage.
9- The first 5 paragraphs of the discussion should be included in the introduction as they explain the reason why this study was done.
10- The article needs english proofreading.
Author Response
Authors are grateful for evaluation and valuable comments that helped us to improve our paper.
- It has been corrected.
- We excluded patients who started ADT use more than 3 months after diagnosis to have group of prostate cancer patients with sufficient ADT exposure (short term users were excluded).
- At the beginning of our study, we panned to compare two separate groups of ADT users (GNRH agonists and antiandrogens). However, after exclusion has been done, we found out that we only have no more than 45 cases of only antiandrogen users and comparison between those two groups would not be possible. So we have decided to eliminate only antiandrogen users at the end.
- We chose to list inclusion criteria in every single paragraph in terms of the topic. Inclusion criteria are listed in the lines 66-67, 73-77, 83-85. Exclusion criteria is listed in the lines 85-92.
- Thank you very much for your notice. Indeed it was our mistake. Intervals have been corrected in the table and in the text as well into 4-43, 44-104 and >105 weeks. Intervals in our study were chosen by dividing a scale into categories based on tertiles.
- I has been corrected and simplified.
- Values have been added to the Table 1.
- It has been changed according to your suggestion.
- The introduction has been improved according to your suggestion.
- The article has been checked by native English speaker.
Round 2
Reviewer 2 Report
I hold my original decision.